

# In situ measurements of angular dependent light scattering by aerosols over the contiguous United States

W. Reed Espinosa[1,2], J. Vanderlei Martins[1,2], Lorraine A. Remer[1,2], Anin Puthukkudy[1,2], Daniel Orozco[1,2], and Gergely Dolgos[1,2,3]

[1]Department of Physics, University of Maryland Baltimore County, 1000 Hilltop Circle, Baltimore, MD 21250, USA
[2]Joint Center for Earth Systems Technology, University of Maryland Baltimore County, 5523 Research Park DR, Baltimore, MD 21228, USA
[3]Micos Engineering GmbH, Überlandstrasse 129, CH-8600, Dübendorf, Switzerland

*Correspondence to:* W. Reed Espinosa (reedespinosa@umbc.edu)

**Abstract.** This work provides a synopsis of aerosol phase function ($F_{11}$) and polarized phase function ($-F_{12}/F_{11}$) measurements made by the Polarized Imaging Nephelometer (PI-Neph) during the Studies of Emissions and Atmospheric Composition, Clouds and Climate Coupling by Regional Surveys (SEAC[4]RS) and the Deep Convection Clouds and Chemistry (DC3) field campaigns. In order to more easily explore this extensive dataset, an aerosol classification scheme is developed that identifies the different aerosol types measured during the deployments. This scheme makes use of ancillary data that includes trace gases, chemical composition, aerodynamic particle size and geographic location, all independent of PI-Neph measurements. The PI-Neph measurements are then grouped according to their ancillary data classifications and the resulting scattering patterns are examined in detail. These results represent the first published airborne measurements of $F_{11}$ and $-F_{12}/F_{11}$ for many common aerosol types. We then explore whether PI-Neph light-scattering measurements alone are sufficient to reconstruct the results of this ancillary data classification algorithm. Principal component analysis (PCA) is used to reduce the dimensionality of the multi-angle PI-Neph scattering data and the individual measurements are examined as a function of ancillary data classification. Clear clustering is observed in the PCA score space, corresponding to the ancillary classification results, suggesting that indeed a strong link exists between the angular scattering measurements and the aerosol type or composition. Two techniques are used to quantify the degree of clustering and it is found that in most case the results of the ancillary data classification can be predicted from PI-Neph measurements alone with better than 85% recall. This result both emphasizes the validity of the ancillary data classification as well as the PI-Neph's ability to distinguish common aerosol types without additional information.

## 1 Introduction

Atmospheric particulates can directly affect Earth's energy balance through the scattering and absorption of light (Bellouin et al., 2005), as well as serve as cloud condensation nuclei (CCN), leading to changes in cloud properties and precipitation patterns (Rosenfeld et al., 2008). The most recent IPCC assessment identifies both aerosols' direct and indirect effects as the two largest uncertainties of all anthropogenic radiative forcing components (Stocker, 2014). Spaced based remote sensing and model simulations yield aerosol data with global coverage but the assumptions and validation of these techniques requires



additional information. A wide range of methods exist to provide these additional constraints, with two of the most accurate approaches being airborne in situ measurements and ground-based remote sensing (Remer et al., 1997). The latter approach, specifically in the form of a global array of sun photometers known as the Aerosol Robotic Network (AERONET), has been used extensively to characterize aerosols and validate space-based observations. While this method is remarkably accurate, it

inevitably contains limitations that many situ measurement techniques do not share. For example, the parameters provided by the AERONET retrieval are limited at 440nm optical depths larger than 0.4 and the technique can not provide the vertically resolved aerosol properties that can be obtained by airborne in situ instrumentation.

The limited number angles sampled by typical passive satellite sensors, along with the need to correct for surface reflectance, means that inversions designed to retrieve aerosol properties from these measurements require significant assumptions about

the aerosol in question. These assumptions frequently take the form of a set of aerosol types (ex. desert dust, biomass burning, urban emissions, etc.) with predefined characteristics that can be used to estimate the optical properties of the aerosol. While current algorithms are adequate to retrieve AOD and a few other parameters, the results from a wide range studies have suggested that there may still be room for significant improvements in the aerosol properties assumed in spaced based remote sensing retrievals. For example, localized tests using modified, more locally appropriate aerosol models have shown significant

improvements in comparisons with AERONET derived AOD as well as the ability to increase spatial resolution with little cost to retrieval accuracy (Bilal et al., 2013; Lee et al., 2012; Wong et al., 2011).

Models used to calculate aerosol forcing and to estimate climate change also rely on assumptions regarding aerosol optical properties. In fact, comparison of nine widely used aerosol forcing models found that the greatest diversity in model estimates of forcing were not in the representation of aerosol loading by the models, but in the forcing efficiency, the forcing per unit

of loading (Schulz et al., 2006). The forcing efficiency is affected by wide-ranging values of aerosol optical properties found in the models (Schulz et al., 2006). A revisit of this model comparison, now involving 16 models, published seven years later found no narrowing of model diversity in estimates of aerosol radiative effects and forcing, and significant diversity when analyzed individual aerosol components (Myhre et al., 2013). Again, the reason was traced to significant range of values for factors such as forcing efficiency that stem from lack of constraints in basic aerosol intrinsic properties (Myhre et al., 2013).

The aerosol characteristics used in passive remote sensing algorithms and climate modeling have been primarily based on inversions of AERONET sky radiance measurements that produce values of total column ambient aerosol properties (Dubovik et al., 2000; Dubovik and King, 2000). Constructing aerosol models from these inversion data requires calculating statistics of the quantities for different groupings of the data corresponding to different aerosol types or classifications. Classifications can be identified using a priori knowledge of dominant aerosol types in different locations (Remer et al., 1997; Remer and Kaufman,

1998; Dubovik et al., 2002; Giles et al., 2012), or by using advanced statistical methods such as cluster analysis (Omar et al., 2005; Levy et al., 2007; Wu and Zeng, 2014), recently utilizing Mahalanobis distances (Russell et al., 2014; Hamill et al., 2016). These techniques have also been applied to other high-quality aerosol remote sensing data sets such as High Spectral Resolution Lidar (HSRL) or Multiangle Imaging SpectroRadiometer (MISR) to classify aerosol into dominant types and to derive aerosol models for each type (Burton et al., 2012; Kahn and Gaitley, 2015). The advantage of using remote sensing

data sets to construct aerosol models for remote sensing or climate applications is to have a set of aerosol models producing





a radiance at the top of the atmosphere consistent with the radiance a satellite would measure or that affects the planetary energy balance. The disadvantage of using such data sets to construct aerosol models is that detailed particle information is lost due to ambiguities concerning humidification and height of the particles, and such models cannot be easily linked to particle composition.

In situ measurements of aerosol's optical properties have the potential to provide additional information, like aerosol vertical dependence and particle hygroscopicity, while maintaining an optical consistency comparable to remote sensors. In practice, though, the majority of in situ optical measurements have been limited to parameters, like total scattering, absorption and extinction, which lack the angular information required for a compressive aerosol model. Polar nephelometers have occasionally been used to measured the angular dependence of light scattered by ambient aerosols but measurements by these instruments

above ground level has been extremely limited. The first published measurements from an airborne polar nephelometer were made in the early 1970's (Grams et al., 1975). These measurements were quite advanced for the time but the results presented by the authors are limited to only two phase function averages and lack the context of modern aerosol science. The next polar nephelometer, designed specifically for the measurement of aerosols, to be flown aboard an aircraft was the Polarized Imaging Nephelometer (PI-Neph), which has been active since the fall of 2012 (Dolgos and Martins, 2014). This instrument has sam-

pled extensively during recent field campaigns and, through the use of modern technology, is able to make significantly more advanced measurements than early polar nephelometers. In addition to the two aerosol instruments, a third polar nephelometer originally designed for the measurement of cirrus clouds (Gayet et al., 1997a, b), has been flown extensively since the 1990's. Recently, this instrument has been modified to allow for measurements of some aerosol particles as well as ice crystals, specifically those with diameters larger than 1μm and sufficiently higher number concentrations. The results of these measurements

are presented by Shcherbakov et al. (2016), who performed principal component analysis on the light scattering data to explore the properties of aerosols found in volcanic degassing plumes.

This work explores recent airborne measurements made by the PI-Neph and constitutes the first comprehensive analysis of angular dependent light scattering measurements made in situ from aboard an aircraft on common atmospheric aerosol types. The analysis focuses on measurements made during the Studies of Emissions and Atmospheric Composition, Clouds and

Climate Coupling by Regional Surveys (SEAC[4]RS) and the Deep Convection Clouds and Chemistry (DC3) field campaigns. The light scattering data include both phase function ($F_{11}$) and linear degree of polarization ($-F_{12}/F_{11}$) measurements ranging from 4° to 174° in scattering angle. These measurements are separated into 2390 different averaging periods for which stable, high quality data were available. A classification strategy is then developed for the SEAC[4]RS data to estimate the dominate aerosol type for each case, making use of ancillary trace gas measurements, aerodynamic size distributions and aerosol

composition measurements. The data from the DC3 campaign, which has significantly different objectives from SEAC[4]RS, is classified according to the region where the sample was taken. Principal component analysis (PCA) is then applied to the PI-Neph averages to confirm the validity of the ancillary data classification scheme in a light scattering context, and to explore whether there is enough information in the PI-Neph measurements to classify aerosol types from light scattering alone.





## 2   Methodology

The dataset used in this work is built from measurements made aboard the DC-8 aircraft during the DC3 and SEAC[4]RS field campaigns, with a focus on measurements made by the PI-Neph airborne polar nephelometer. Time averages are performed on the raw PI-Neph data and the resulting cases are grouped into one of eight predefined aerosol categories according to a novel aerosol classification scheme. This classification scheme makes use of measurements that are independent of aerosol light scattering, including particle composition, aerodynamic size distribution and gas concentrations.

### 2.1   DC3 and SEAC[4]RS field campaigns

The DC3 field campaign took place over the central United States in May and June of 2012. The experiment was designed to shed new light on storm dynamics and the effect of convective systems on the chemical composition of the troposphere (Barth et al., 2015). Over the course of the experiment three aircraft flew dozens of flights with a combined payload of over sixty different instruments providing remote sensing and in situ measurements of a wide range of trace gases, aerosols properties and meteorological parameters. The majority of flights focused on one of three study regions: northeastern Colorado (CO), northern Alabama (AL) and a region comprising northern Texas and southern Oklahoma (TX/OK) (Barth et al., 2015). The NASA DC-8 aircraft was typically used to sample storm inflow, while the National Center for Atmospheric Research (NCAR) Gulfstream V (GV) and the German Deutsches Zentrum für Luft- und Raum- fahrt (DLR) Falcon sampled the outflow regions. While the DC-8 did occasionally sample convective system outflow, PI-Neph data corresponding to these periods were infrequent and highly variable. In order to simplify the analysis no PI-Neph measurements corresponding to storm outflow in DC3 have been included in this work.

In August of 2013 the associated SEAC[4]RS campaign begin its two-month long deployment, with flights covering much of the contiguous United States (CONUS). The campaign targeted a variety of atmospheric phenomena including the role of convection in the distribution of aerosols and gases within the troposphere, the climatic and meteorological effects of biomass burning and anthropogenic emissions and the calibration and validation of satellite data. The aircraft supporting the airborne portion of the experiment included the NASA ER-2, the NASA DC-8, and SPEC Inc. Learjet. These three platforms flew a total of 57 different flights and had a combined instrument payload very similar to DC3. The PI-Neph sampled from the DC-8 aircraft in both DC3 and SEAC[4]RS, and the data used in this work relies primarily on measurements made aboard this platform. A detailed description of the SEAC[4]RS scientific goals, aircraft and instrumentation, as well as the corresponding implementation can be found in Toon et al. (2016).

### 2.2   Instrumentation

The PI-Neph uses a wide field of view imaging system to measure the angular dependence of light scattered by aerosols and the surrounding gases (Dolgos and Martins, 2014). The sample is illuminated by a high-powered continuous wave laser that is folded on one end of the sample chamber. This folding of the beam allows the forward and backward scattering angles to be captured in volumes that are physically adjacent and parallel to each other, reducing the overall length of the instrument.





While this design permits a physical footprint small enough to fit inside the limited space of an aircraft cabin it also introduces the need for separate calibrations of both the forward and backward scattering angles (Dolgos and Martins, 2014). In DC3 the PI-Neph utilized only one laser operating at 532nm, but two additional lasers were incorporated into the instrument prior to SEAC[4]RS, adding measurements at 473nm and 671nm. The scattered laser light is imaged with a wide angle refractive lens

and charge-coupled device (CCD) camera. A single measurement is composed of two sequential images, one for each of two approximately orthogonal linear polarization states of the laser.

If the scattering medium is assumed to be macroscopically isotropic and symmetric the scattering matrix elements $F_{13}$ and $F_{14}$ do not contribute to the total scattered signal and the resulting pair of image intensities allows for direct measurements of $F_{11}(\theta)$ as well as $F_{12}(\theta)$, with $\theta$ representing the zenith scattering angle (azimuthal symmetry is implied by the assumption of

a macroscopically isotropic and symmetric medium). The incorporation of calibration data derived from molecular scatterers ($CO_2$ and $N_2$) that are well characterized (Anderson et al., 1996; Young, 1980) allows for an angular dependent calibration that produces direct measurements of absolute phase function in known units (ex. $\mathrm{Mm}^{-1}\mathrm{sr}^{-1}$), free from truncation error. Assumptions regarding the relative scattering contribution of the extreme angles can then be used to estimate total integrated scattering ($\beta_{scat}$) from the truncated measurements of absolute phase function.

The angular resolution of the measurement is limited by the resolution of the CCD camera, as well as the size of the camera's aperture. The resulting raw resolution varies as a function of scattering angle ($0.1° < \Delta\theta < 1°$) but the final results are always binned to one degree. The angular range of the instrument is limited by stray light emanating from the points where the laser beam enters and exits the sample chamber. Stray light can vary significantly with instrument alignment, but the measurement bounds were typically 4° to 174° in SEAC[4]RS, and 5° to 170° in DC3. The final products are then reported at standard temper-

ature and pressure, with the Rayleigh scattering contribution from the surrounding gases subtracted. Additionally, when phase functions are normalized in this work they are represented by $\tilde{F}_{11}$ and are scaled such that $\tilde{F}_{11}(30°) = 1$. This normalization strategy avoids the truncation errors produced by other schemes that are based on the integral of $F_{11}$ over the full range of zenith scattering angles.

In both SEAC[4]RS and DC3, ambient air was provided to the PI-Neph through the NASA Langley Aerosol Research Group

Experiment's (LARGE) shrouded diffuser inlet (McNaughton et al., 2007), which sampled isokinetically. A flow of 20 liters per minute was maintained through the PI-Neph's 10-liter sample chamber, leading to an aerosol exchange time on the order of 30 seconds. The typical raw sampling rate of the instrument was 45 seconds in SEAC[4]RS and 11 seconds in DC3 but all data shown in this work are averages composed of multiple raw measurements. The incorporation of the two additional measurement wavelengths gives rise to the longer PI-Neph sampling time in SEAC[4]RS.

The PI-Neph's sample was conditioned with a temperature-controlled drier that heated the incoming ambient air to a temperature of 35°C and, in almost all cases, kept the relative humidity of the sample below 40%. When heating the sample aerosol, it is possible to evaporate volatile compounds and significantly perturb the aerosol properties (Shingler et al., 2016), but this effect is not believed to have played a consequential role on PI-Neph measurements made during DC3 and SEAC[4]RS. In order to better understand the biases produced by the evaporation of volatile compounds PI-Neph total scattering measurements were

compared with scattering data from an integrating nephelometer (model 3563, TSI Inc., St. Paul, MN, USA) using a nafion





drier that did not require sample heating. A strong correlation was observed between the two instruments ($R^2 > 0.995$) and no decrease in PI-Neph scattering, relative to integrating nephelometer scattering, was observed during periods corresponding to large temperature gradients between the PI-Neph's sample chamber and the ambient air. It should be noted that the PI-Neph and integrating nephelometer sampled from the same inlet so the results of this comparison do not preclude effects from other

heating mechanism like ram heating (adiabatic heating associated with decelerating flow) and heat exchange with the aircraft cabin inside the sample tubing (Wendisch et al., 2004; Baumgardner et al., 2011).

In this work the Particle Analysis by Laser Mass Spectrometry (PALMS) instrument was used to aid in the identification of aerosols containing significant amounts of mineral dust. PALMS uses a strong ultra-violet laser pulse to ablate particles, the ionized fragments of which are then passed through a time-of-flight mass spectrometer (Thomson et al., 2000). The quantity of

10 alumina and aluminosilicates is then used to identify mineral dust particles (Lee et al., 2002) and the fraction of these particles is reported over five-minute intervals. In this work dust aerosols are also classified using information regarding aerodynamic particle size. An aerodynamic particle sizer (APS model 3321, TSI Inc., St. Paul, MN, USA), measuring particle time-of-flight inside an accelerating air flow, was used to obtain these measurements. APS measurements were made at ambient humidities during SEAC$^4$RS and the results were reported in 14 log spaced bins with midpoint diameters ranging from 563nm to 6.31μm.

Trace gas concentrations are used to identify air masses corresponding to urban, biogenic and biomass burning emissions. Carbon monoxide volume mixing ratios were obtained with the Differential Absorption Carbon Monoxide Monitor (DACOM; Fried et al. (2008)). Measurements of nitrogen dioxide (NO$_2$) were made by NOAA's NOyO3 instrument using the UV-LED photolysis-chemiluminescence technique (Pollack et al., 2010; Ryerson et al., 2000). The University of Innsbruck's High-Temperature Proton-Transfer-Reaction Mass Spectrometer (HT-PTR-MS; Mikoviny et al. (2010)) was used to quantify the

mixing ratio of the remaining gas species, specifically acetonitrile (CH$_3$CN), isoprene (C$_5$H$_8$) and monoterpenes (C$_{10}$H$_{16}$).

### 2.3 Averaging of PI-Neph measurements

The PI-Neph made more than ten thousand raw measurements over 163 flight hours during the SEAC$^4$RS campaign and almost forty thousand raw measurements over 116 hours during DC3. A significant fraction of these measurements occurred at very low aerosol concentrations, typically during high altitude transit legs of the flights, when noise can overwhelm the scattering

signal. Additionally, examination of the measurement data showed that, while the aerosol concentrations often varied quite quickly, the values of $\tilde{F}_{11}$ and $-F_{12}/F_{11}$ were generally stable over much longer periods. These facts motivated the decision to perform averages on the raw PI-Neph data over periods corresponding to several measurements.

The averaging scheme was designed to both reduce random noise as well as eliminate periods of very low scattering where systematic (i.e. temporally correlated) sources of error are significant. Only raw PI-Neph measurements corresponding to high

aerosol concentrations and relatively stable optical properties were included in the averaging scheme. Specifically, only measurement periods where the total scattering was consistently above 10Mm$^{-1}$ and the change in integrated scattering between two adjacent raw measurements was less than 15 % were considered. If insufficient or unstable scattering led to the removal of a raw data point the relevant average was discarded and a new potential averaging window was started (i.e. all averages are composed of consecutive data points). An averaging period was concluded once at least three raw measurements were included





and the sum integrated scattering values of each individual data point summed to greater than $200 \mathrm{Mm}^{-1}$. The averaging periods were derived from the $532\mathrm{nm}$ SEAC[4]RS products to maximize consistency in the averaging procedure between the two campaigns.

The process described above resulted in 573 averages in SEAC[4]RS and 1817 averages from DC3. The mean average time in SEAC[4]RS was 152 seconds while the mean averaging time during DC3 was 67.6 seconds. 93% SEAC[4]RS and 67% of the DC3 averages were made on 5 raw data points or less. The smaller quantity of cases (and longer mean averaging times) in the case of SEAC[4]RS is primarily due to the reduced time resolution associated with the three-wavelength measurement.

## 3  Aerosol classification using ancillary data

An aerosol classification scheme was developed to estimate the dominate source of each aerosol by focusing on the airmass associated with each PI-Neph average, using ancillary data that include measurements of gases, aerosol composition and physical properties as well as aircraft location. Aerosol optical properties were intentionally omitted from all classification metrics to ensure independence between the classification scheme and the scattering features measured by the PI-Neph. As convective systems have the potential to significantly influence aerosol properties (Jeong and Li, 2010; Eck et al., 2014; Corr et al., 2016) different classification schemes were applied to the DC3 dataset (near convective systems) and the SEAC[4]RS data set (generally far from convective systems). The SEAC[4]RS data was subdivided into five categories corresponding to dust, biogenic, urban, biomass burning (BB) emissions and unclassified samples. This classification utilized measurements of particle chemical composition from the PALMS instrument, the aerodynamic size distribution of particles generally associated with the coarse mode and a range of trace gases. The scheme developed to categorize the DC3 data was based on aircraft location relative to three storm domains outlined in the DC3 science objectives (Barth et al., 2015) as well as the presence of convective systems over the course of the corresponding flight. Both the DC3 and SEAC[4]RS classification schemes only allow one aerosol type to be assigned to a given PI-Neph sample.

The SEAC[4]RS dust classification requires that the PALMS instrument identify at least 15% of the measured particles as mineral dust. It was found that the PALMS algorithm would often classify a significant fraction of particles as mineral dust, even when particle size distribution measurements showed no significant coarse mode. In order to exclude these cases and align our dust classification with more traditional dust aerosol we imposed a set of requirements on the coarse mode of the aerodynamic size distributions. Specifically, the volume concentration measured by the APS (which is insensitive to particles below $500\mathrm{nm}$) must exceed $2\mathrm{\mu m}^3/\mathrm{cm}^3$ and have an effective radius greater than $750\mathrm{nm}$. This constraint on the aerodynamic particle size distribution removed several cases where no obvious source of dust could be identified.

If the dust category was not selected gas tracers and aircraft altitude were used to screen for the remaining three fine mode dominated types. Shingler et al. (2016) used a threshold of $250\mathrm{pptv}$ acetonitrile—or $250\mathrm{pptb}$ of carbon monoxide if acetonitrile data is unavailable—as an indicator of BB emissions. We have modified this metric to also include cases with acetonitrile values as low as 190 pptv but only if the sum of the volume mixing ratios of isoprene and monoterpenes is less than 40% that of acetonitrile. Since isoprene and monoterpenes are well correlated with biogenic emissions this condition permits



the inclusion of cases with lower BB concentrations, while still avoiding false positives that can potentially be triggered by strong biogenic emissions of acetonitrile. Accordingly, isoprene and monoterpenes are used as gas tracers for the biogenic category, with a biogenic classification occurring when their combined concentrations exceed $2\,\mathrm{ppbv}$. When the previous three categories are not triggered, the aircraft is within or close to the mixing layer (altitude below $3\,\mathrm{km}$) and $NO_2$ concentrations

are greater than $1\,\mathrm{ppbv}$ the urban category is selected. A marine aerosol classification occurring whenever the aircraft was directly above large bodies of water was also examined, but the scattering intensity during almost all corresponding periods was below the PI-Neph's lower limit of detection. The few remaining marine cases, as well as all cases that failed to trigger any other classification, are identified as "unclassified". A decision tree specifying the requirements for each SEAC[4]RS category is shown in Figure 1.

The DC3 campaign had significantly different objectives—namely the study of convective systems—and correspondingly the SEAC[4]RS classification algorithm does not map well to the DC3 dataset. This fact motivated the decision to classify the DC3 samples by study region (CO, AL or TX/OK) as opposed to gas and composition data. A PI-Neph sample was associated with a given study region if the aircraft coordinates were within the corresponding domain and the corresponding flight path was designed to target active storms in the region. Additionally, in order to restrict the assigned cases to storm inflow measurements,

the classification was only applied if the observation was made below $6\,\mathrm{km}$. This constraint produced much more homogeneous aerosol properties for each storm domain by eliminating the cases with highly variable scattering properties found in the higher altitude outflow aerosols.

Table 1 shows the number of cases the ancillary data classification scheme assigned to each category, as well as the number of unique flights containing at least one of the corresponding cases. 70% of the SEAC[4]RS cases received a classification (other

than 'unclassified') while 55% of the DC3 cases were classified. In both campaigns, the majority of the unclassified cases correspond to high altitude transit legs that are generally associated with relatively clean air masses. All categories have cases originating from multiple flight days, increasing the likelihood that a given category average is representative of the typical aerosol properties found in that type.

Figure 2 shows the geographic locations of all classified PI-Neph samples. The extent of the three DC3 study regions can be

seen in the spread of the red, beige and maroon circles corresponding to the CO, TX/OK and AL storms categories, respectively. In the summer months, biogenic emissions often dominate the south eastern United States (SEUS) while the western portion of the country is frequently influenced by wildfire smoke. While the SEAC[4]RS categories are not determined by location, clear patterns emerge that are in strong agreement with these physical expectations.

The California Rim Fire was one of the dominate sources of biomass burning emissions during the SEAC[4]RS deployment.

The fire began on August 17[th] in Stanislaus National Forest, California and continued to burn until after the end of the SEAC[4]RS deployment (Saide et al., 2015). Before the fire was fully extinguished its total burn area had grown to 104,000 $\mathrm{ha}$, making it the third largest fire in California's history (Peterson et al., 2015). The arch of biomass burning cases that is shown in the northwestern portion of Figure 2 and traverses from California into southern Canada represents samples dominated by Rim Fire smoke. Emissions from this fire made up 58% of all classified BB cases. The overwhelming majority of the remaining cases

came from smaller wildfires within the United States, primarily from three fires located in Wyoming, Colorado and Kansas





(Toon et al., 2016). While many agricultural fires were sampled in SEAC$^4$RS these measurements were almost always discarded by the averaging algorithm described in Section 2.3 due to the very short duration, and high variability, of the corresponding measurement.

The 15 points that met the requirements of the dust classification are shown in yellow in Figure 2. This type was only observed in early August over Louisiana and the northern Gulf of Mexico. These cases likely correspond to a transported Saharan Air Layer (SAL) that was present over this region at the start of the campaign. There is strong evidence, based on aerosol concentration and composition, that this airmass was relatively pristine and had not mixed significantly with continental air (Ziemba et al., 2016).

In August and September biogenic emissions are ubiquitous in the south eastern United States. The classification scheme presented here conveyed this fact well, with most of these cases in this region falling under the biogenic category. The second most prevalent category over the SEUS is the urban type. This classification corresponds well to city centers like Houston and Dallas Texas, whose emissions were frequently sampled by the DC-8. Additionally, a large strip of urban cases can be seen around the Ohio River Valley, an area with a very high concentration of fossil fuel based power plants. It is shown in Section 5 that the optical properties of aerosols associated with cities are quite different from the industrial emissions of the Ohio River Valley. The possibility of dividing the urban category into two sub-groups was explored, but the already limited number of cases made this division impractical. It is likely that other datasets, with a larger number of samples corresponding to urban and industrial emissions, can be more easily understood by dividing the urban classification described here into two separated sub-categories that are separated by SO$_2$ concentrations, for example.

## 4 Measurement of phase matrix elements

A robust averaging procedure was applied to all sample averages of $\tilde{F}_{11}$ and $-F_{12}/F_{11}$ data of a given aerosol type to obtain curves that are typical of each category. Figure 3 shows the results of this averaging for all three DC3 storm domains as well as the dust category and the average of all three fine mode dominated aerosols (biogenic, urban and BB) from SEAC$^4$RS.

A progression in both $\tilde{F}_{11}$ and $-F_{12}/F_{11}$ averages is evident as the DC3 storm domain transitions from AL to TX/OK to CO. The increase in forward scattering peak through this sequence suggests an increased scattering contribution from particles in the larger size ranges that direct the bulk of their scattered energy into the forward angles. Models have shown that large convective systems can agitate surface dust, drawing these particles up into the atmosphere and acting as a significant source of dust aerosol (Seigel and van den Heever, 2012; Takemi et al., 2006). The elevated forward scattering peaks are likely driven by increasingly arid surface features, leading to an increased availability of this relatively large dust aerosol (Tulet et al., 2010). Variations in typical storm wind speeds may have also contributed to variations in the quantity of dust that was suspended over a given region.

The same progression is evident in the backscattering angles of the DC3 storm categories, with CO having the strongest backscattering intensities, followed by TX/OK and then AL. In size distribution and refractive index regimes typical of ambient aerosol, this region of the phase function is very sensitive to the diameter of the fine mode particles, suggesting significantly





smaller fine mode particles in the CO inflow than in AL. A fine mode peak shifted toward smaller diameters also produces larger values of $-F_{12}/F_{11}$ at side scattering angles, although this effect is partially moderated by differences in refractive index.

The average $\tilde{F}_{11}$ data, corresponding to the SEAC⁴RS fine mode dominated categories, shows very weak forward scattering and thus suggest relatively few coarse mode particles. The low peak value of $-F_{12}/F_{11}$ observed in this fine mode dominated data is likely driven primarily by fine mode particles with slightly larger diameters than those found in the three DC3 categories. The features of the dust scattering matrix elements, specifically the presence of an extremely strong forward scattering peak, are typical of an aerosol whose scattering properties are dominated by coarse mode particles. While the typical integrating scattering intensity for dust was comparable to other aerosol types, the strong forward scattering peak significantly limits the amount of light scattered at other angles. The combination of this low absolute scattering intensity and systematic instrument noise resulting from stray light may produce significant biases in the dust $\tilde{F}_{11}$ and $-F_{12}/F_{11}$ measurement averages at angles above $90°$.

The averages of the three SEAC⁴RS fine mode cases are examined individually in Figure 4. Visually the averages of the three types produce very similar angular scattering patterns, especially the biogenic and urban averages. The $-F_{12}/F_{11}$ peak was slightly larger on average in the biomass burning cases, with this feature most clearly separating the BB aerosols from the other two types. Additionally, small differences in the shape of $\tilde{F}_{11}$ can also be observed in the biomass burning averages, where the forward and backward scattering peaks are suppressed relative to the other two types.

The variability within a given type's scattering data (not shown) was the highest in the case of the samples associated with urban emissions. Further examination of this variability showed two distinct subgroups, with the conditions around the Ohio River Valley differing significantly from conditions near urban centers. The starkest difference between these two subgroups occurred in the $-F_{12}/F_{11}$ maxima, with significantly higher peaks occurring in measurements made near the Ohio River Valley.

## 5    PCA analysis

It is evident from the results of the previous section that there are differences in the averaged scattering data that agree well with the physical expectations of each aerosol type. While this result is encouraging, the averages alone do not tell us if these differences are characteristic of the majority of samples or are driven by a relatively few extreme cases. The regularity of the geographic patterns observed in Figure 2 does suggest a consistent physical basis for the classification scheme in the majority of SEAC⁴RS cases but it says nothing about the DC3 classification where sample location is already the primary classification metric. Additionally, as none of the properties used by the SEAC⁴RS classification scheme are directly related to aerosol optical measurements, it is possible that the geographic distributions observed capture patterns in features of the air masses that are not reflected in the optical properties of the corresponding aerosol populations. In order to confidently say that the majority of cases have aerosol optical properties that are clearly characteristic of the corresponding type we must examine PI-Neph measurements on the level of individual cases. Unfortunately, the subtle differences between many of the





scattering measurements and the high dimensionality of the data set complicates a direct analysis of the relevant features. In the following section, this analysis is simplified by reducing the dimensionality of the PI-Neph measurements with principal component analysis. This approach leads to a clear picture of the type-driven clustering of cases that occurs in scattering element space as well as permits easy identification of the features that are characteristic of each aerosol category.

Principal component analysis was performed on all PI-Neph measurement averages to simplify the scattering data and more easily explore its relationship with the classification categories. Intuitively, PCA transforms the data to a new coordinate system in such a way that the greatest variance lies along the first coordinate, the next largest variance lies along the second coordinate and so forth (Jolliffe, 2002). This process allows most of the data set's variance to be captured in the first few principal components and in turn the dimensionality of the measurement can be significantly reduced while still maintaining the bulk of

the original information content. Mathematically, the basis vectors of this new coordinate system are the eigenvectors of the data's covariance matrix. In this work the orthonormal basis vectors describing this new coordinate system (i.e. the normalized eigenvectors of the covariance matrix) are referred to as 'loadings', and the basis vector coefficients required to represent each data point are referred to as 'scores'. It is important to note that PCA is an unsupervised technique and the results are therefore independent of any hypothesis regarding the data, including the ancillary data classification scheme.

PCA was performed on all $532$nm PI-Neph averages from the combined SEAC[4]RS and DC3 datasets (2,334 samples) simultaneously. While the data from the two campaigns was merged, the unpolarized and polarized measurements were kept separate in the final analysis (i.e. the PCA routine was run twice, once for $\tilde{F}_{11}$ and again for the $-F_{12}/F_{11}$ dataset). Individual $\tilde{F}_{11}$ measurements can often span several orders of magnitude between the forward scattering peak and side scattering angles. To prevent the first few principal component loadings from being dominated by the large absolute variations in forward scattering

intensity the analysis was performed on the natural logarithm of the $\tilde{F}_{11}$ values. This transformation produces a set of principal components where the first component, for example, explains the largest possible *relative* variance in data (Shcherbakov et al., 2016). No transformation was applied to the $-F_{12}/F_{11}$ measurements. The angular range of the final inputs to the PCA routine was $5°$ to $170°$ in the case of both the $\tilde{F}_{11}$ and the $-F_{12}/F_{11}$ datasets. This range corresponds to the angles where data was present during all measurement periods over both campaigns. Data points where instrument noise produced non-physical (i.e.

$F_{11}(\theta) < 0$ or $|F_{12}(\theta)/F_{11}(\theta)| > 1$) values were excluded from the analysis.

The decision to treat the intensity and polarization information separately when performing the PCA was based on two factors. The first stems from the fact that most modern measurements of the optical properties of atmospheric aerosol are polarization insensitive. Isolating the polarization information permits conclusions that are more applicable to polarization insensitive instrumentation while simultaneously helping to illuminate the potential benefits of adding polarization capabilities

to future instrumentation. The second factor results from the fact that PI-Neph data contains sources of systematic noise that are strongly correlated over time and scattering angle but is very weekly correlated between $\tilde{F}_{11}$ and $-F_{12}/F_{11}$. By separating these datasets, the variability in the data corresponding to these systematic artifacts can be more effectively isolated, allowing the remaining components to more accurately capture the physical variation among the samples. This hypothesis is supported by the fact that some of the loadings closely matched the angular error correlations known to result from certain instrumental

artifacts. Additionally, a significantly reduced separation of aerosol types in PCA score space was observed when PCA was





performed on the intensity and polarized measurements simultaneously. This observation was consistent regardless of the relative weights applied to the $\tilde{F}_{11}$ and $-F_{12}/F_{11}$ variances.

## 5.1 PCA loadings and scores

The PCA loadings derived from the combined dataset of all DC3 and SEAC[4]RS measurements are shown in Figure 5. The first three $\tilde{F}_{11}$ components explained 84% of the total variance in the $\tilde{F}_{11}$ data, while the corresponding three $-F_{12}/F_{11}$ components were able to explain 65% of the variance in the $-F_{12}/F_{11}$ measurements. The second $-F_{12}/F_{11}$ loading closely matched a known measurement artifact that is driven by small variations in PI-Neph laser power over the course a given measurement. Similarly, the fourth $\tilde{F}_{11}$ loading (not shown) matched a known artifact produced by relative drifts in the calibration of the forward and backward scattering angles, often driven by fouling of the beam folding mirror inside the PI-Neph chamber.

A 3D scatter plot of the scores from the first two $\tilde{F}_{11}$ principal components and the first $-F_{12}/F_{11}$ component is shown in Figure 6. The points are colored according to the classified results of the aerosol typing algorithm described in Section 3 (unclassified points are excluded for clarity). A simple physical interpretation of the individual principal components is not readily apparent but strong clustering of the points as a function of aerosol type is evident. The grouping of aerosol types purely by optical means suggests that the optically independent ancillary data classification algorithm is capable of capturing significant underlying commonalities in particle properties that extend beyond the metrics directly used by the algorithm itself. Conversely, this result shows that aerosol types can be identified very reliably using only PI-Neph light scattering measurements.

In order to quantify the level of clustering by aerosol types in PCA score space the success of two different prediction algorithms are evaluated against the data. These algorithms attempt to predict the results of the ancillary data classification using only the PCA scores of the corresponding PI-Neph average by exploiting the clustering of each aerosol type. The dimensionality of the original data (166 angles in both $\tilde{F}_{11}$ and $F_{11}/F_{12}$) is often significantly higher than the number of measurements available for a given aerosol type. If all principal component scores were used in the prediction scheme many aerosol types could be identified with very high fidelity based only on the noise "fingerprints" of their individual measurements. Reducing the dimensionality of data down to only a few key variables (i.e. the first few principal components) forces the classification to rely primarily on physical features of the measured aerosol that are common to all samples of that type. Sections 5.2 and 5.3 describe two predictions schemes used to quantify the degree of separation between each of these populations, while simultaneously exploring the distinguishing optical characteristics of each category.

## 5.2 Identifying types by Mahalanobis distance

The first of the two prediction schemes estimates the ancillary data classification of a given sample based on the Mahalanobis distance between that sample's point in PCA score space and the corresponding clusters of points defined by the classification algorithm. Mathematically, the Mahalanobis distance $D_M(\boldsymbol{x})$ of a given point $\boldsymbol{x} = (x_1, x_2, ..., x_n)^T$ from the mean of a cluster





of points $\boldsymbol{\mu} = (\mu_1, \mu_2, ..., \mu_n)^T$ is defined by

$$D_M(\boldsymbol{x}) = \sqrt{(\boldsymbol{x} - \boldsymbol{\mu})^T S^{-1} (\boldsymbol{x} - \boldsymbol{\mu})} \tag{1}$$

where $S$ represents the covariance matrix of all points in the cluster and the superscript $T$ represents the transpose of the corresponding vector (McLachlan, 2004). Intuitively, the Mahalanobis distance provides a metric of the separation between

a test point and a cluster of points, scaled by the dispersion of the cluster along the axis passing though the test point and the center of cluster. The use of Mahalanobis distance in this prediction technique permits the algorithm to take the size and shape of each cluster into account when attempting to discriminate between types and prevents classification types with more loosely bound clusters from being "disadvantaged" when evaluating the distance to a given point from the cluster in question. For example, a point lying halfway between (in euclidean space) the dust and AL storm clusters would have a much shorter

Mahalanobis distance to the center of the more disperse dust cluster.

     Specifically, the predicted ancillary data classification of a given location in PCA score space is equal to the type whose cluster of points has the shortest Mahalanobis distance to the point in question. This process effectively uses the Mahalanobis distance to divide the PCA score space up into several non-overlapping regions, with each region corresponding to a given classification prediction. The classification boundaries of these regions are recalculated for each measured data point with the

relevant test point excluded (i.e. $\boldsymbol{x}$ is never included in the calculation of $\boldsymbol{\mu}$ or $T$ when determining $D_M(\boldsymbol{x})$) to ensure that the prediction is made using only information that is independent of the measurement in question. All points that were identified as urban emission by the ancillary data classification were excluded from this prediction scheme due to the limited number of data points and large variability in PCA scores. Additionally, all unclassified points were excluded as no consistent patterns are observed in their PCA scores, leaving six remaining categories for which prediction could be made.

The PCA score space, in which the classification procedure described above is carried out, can be made up of any arbitrary combination of principal component scores. In this section the Mahalanobis distance prediction technique is applied to five different combinations of the first three $\tilde{F}_{11}$ PCA scores and first $-F_{12}/F_{11}$ scores. All other PCA score were excluded from this analysis either because of clear influences from known instrument artifacts or due to their inability to explain a significant portion of the data's variance. The recall (fraction of cases of a given type that are correctly identified) is then calculated for all

aerosol types and combinations of PCA scores. These recall values are then used to asses the ability of this separation technique to predict the ancillary data classification of the PCA scores derived from PI-Neph light scattering measurements.

     The resulting recall values for five different combinations of PCA scores is shown in Table 2. If a sufficient number of principal component scores are considered recall is generally quite high, often exceeding 85%. The fact that such a high proportion of individual cases can be correctly identified is surprising considering the very small differences observed among

the $\tilde{F}_{11}$ and $-F_{12}/F_{11}$ averages shown in Figures 3 and 4. This result reinforces the validity of the ancillary data classification scheme. Moreover, it demonstrates the potential power of the PCA technique when applied to light scattering measurements with high angular resolution and range to distinguish aerosol type without the need for ancillary data.

     The improvements in recall, resulting from the inclusion of additional components, provides a measure of their relative importance when attempting to distinguish aerosol types. In the biomass burning and CO storm cases the addition of the





first $-F_{12}/F_{11}$ scores always results in an increase in recall by over 10%, suggesting $-F_{12}/F_{11}$ can play an important role in correctly identifying these types. Conversely, the TX/OK and AL storm cases showed no meaningful improvement in the prediction ability of the Mahalanobis distance algorithm when $-F_{12}/F_{11}$ scores were incorporated. Similar conclusions can be made regarding the different $\tilde{F}_{11}$ components. For example, the biogenic recall is always significantly improved with the

addition of the second and third $\tilde{F}_{11}$ principal components.

The Mahalanobis distance based prediction scheme is significantly less successful on the AL storms than the other aerosol types. In order to better understand this discrepancy, we examine the incorrectly classified cases in more detail. The confusion matrix detailing the prediction scheme's performance for the case where all four principal components are used is listed in Table 3. The rows of this matrix represent instances of the actual ancillary data classification while the columns show

the corresponding number of cases predicted by Mahalanobis distance scheme. It is apparent that the Mahalanobis distance approach has significant difficulty discriminating the AL storm cases from the biogenic cases. This result is not surprising given that the AL storm domain corresponds to a region that was dominated by biogenic emissions during SEAC[4]RS. It is likely that some of these SEAC[4]RS biogenic cases were even measured within the vicinity of strong convective systems, further blurring the boundary between these two types. Additionally, as this is a relatively wet and vegetated region, dust emissions that are

driven by the strong winds associated with convective systems are expected to be significantly less than the other two storm domains.

### 5.3   Identifying types with a dividing plane

The Mahalanobis distance technique effectively identified types that are surrounded by other clusters, but clusters lying on the edge of the PCA score space can potentially be identified more accurately using other techniques. In this section, a plane

is used to divide three dimensional PCA score spaces into two regions, representing positive and negative predictions of a given ancillary data classification. The location of this separating plane is unique to each aerosol type and is chosen to produce the highest quality predictions possible. This technique proves to have stronger predictive power than the Mahalanobis distance technique for several aerosol types, while simultaneously providing a more intuitive picture of the characteristic optical features of a given classification. It also allows for the inclusion of the urban and unclassified points that were discarded in the

Mahalanobis distance prediction scheme.

In this prediction scheme the distance of a given point from the dividing plane strongly corresponds to the likelihood of this point being a member of the relevant aerosol type. Similarly, light scattering features characteristic of the aerosol type in question can be identified by examining the basis vector that corresponds to the line normal to the separating plane in PCA score space. Since the direction of this normal line is determined by a plane, the scattering features corresponding to this direction

in PCA score space are always the same, regardless of the location of the point in question. This fact results from the use of a plane to separate the aerosol types, and is not true of the Mahalanobis distance technique where the classification boundaries are much more complex. For example, in the Mahalanobis distance classification scheme, two points that are diametrically opposite the center of a cluster will require opposite changes in their scattering patterns to increase their probability of being associated with the relevant cluster. It should be emphasized though that the separating plane technique only produces a binary




classification (the test point either is or is not predicted to be of the relevant aerosol type) and the method is ineffective at identifying points in clusters that are surrounded by other clusters.

The dust, biomass burning and CO storms clusters are especially well-suited for the separating plane technique as their principal component scores lie on the outer edge of the other datasets in most dimensions. The technique was applied to each of these aerosol types, once using all three $\tilde{F}_{11}$ principal component scores and again using only the first two of these scores as well as the first $-F_{12}/F_{11}$ score (i.e. the scores plotted in Figure 6). Unless otherwise stated the separating plane was chosen to divide the relevant aerosol type from all other aerosol types, including the 'unclassified' samples.

When using the Mahalanobis distance based technique each point was only assigned to one category so false positives in one aerosol type resulted in a reduction of the recall value in another type. In the separating plane technique, each aerosol type is treated as binary classification problem that is independent of the other airmass types. Therefore, the percentage of cases of a given type that were correctly classified is an insufficient metric as a plane chosen infinitely far from the origin will always result in perfect recall. In order to address this issue, we also make use of the true negative rate (TNR), which is defined as the proportion of cases correctly predicted to not belong to the relevant type out of the total number cases that are not of the relevant type. For each aerosol type the location and orientation of the separating plane was chosen to maximize the product of recall and $TNR$. This metric—the fraction of cases of the relevant type that were correctly predicted times the fraction of cases not of the relevant type that were correctly predicted—takes into account both the sensitivity of the prediction as well as its ability to exclude cases corresponding to other types.

To further examine the technique's ability to separate different aerosols, comparisons were made between a given category and subsets of other categories that only contain types with very similar aerosol properties. The first of these comparisons was made between the biomass burning samples and all other fine mode dominate aerosol types. The fine mode dominate aerosols used include the biomass burning, biogenic, urban/industrial types as well as convective storm inflow from the Alabama domain. These categories were chosen based on optical and aerodynamic side distribution measurements as well as retrievals of PI-Neph data (Espinosa et al., 2017) that showed these categories typically had volume distributions that were dominated by fine mode particles.

Additionally, an attempt was made to separate all the CO storm cases from the AL storm cases. This comparison serves both to exemplify the significance of the differences between the storm domains as well as better clarify the continuum on which the light scattering properties of all DC3 storm domains can be projected.

Table 4 shows the resulting recall and TNR values from the separating plane prediction technique. In most cases the algorithm can predict the classification correctly as well as rejecting cases that are not of the relevant type with better than 90% accuracy. The last column of Table 4 contains a parameter quantifying the role of the first principal component of $-F_{12}/F_{11}$ in identifying the corresponding aerosol type. This value $F_{12}:PROJ$ corresponds to the projection of the unit vector normal to the separating plane, pointing in the direction of the desired classification, onto the axis corresponding to the scores of the first principal component of $-F_{12}/F_{11}$. An absolute value of $F_{12}:PROJ$ approaching unity indicates that the separation is completely determined by $-F_{12}/F_{11}$, while values approaching zero indicate no sensitivity in the first principal component of $-F_{12}/F_{11}$





to the corresponding type. The sign of $F_{12} : PROJ$ indicates whether "more" or "less" of the first principal component of $-F_{12}/F_{11}$ is indicative of the type in question.

In all cases where the $-F_{12}/F_{11}$ scores are included the algorithm predicts the classification correctly with better than 90% accuracy and, with the exception of the CO storms, it shows equivalent skill rejecting cases that are not of the relevant type.

The predictive accuracy of the scheme when using the third $F_{11}$ component is similar, except in the case of the BB samples. This is consistent with the results of the Mahalanobis distance technique of the previous section, where the first $-F_{12}/F_{11}$ component was found to be crucial in obtaining high biomass burning recall. Interestingly, the ability of the dividing plane technique to correctly reject points that were not of the CO storm classification was significantly improved when the first $-F_{12}/F_{11}$ principal component was replaced by the third $F_{11}$ component. This result is somewhat contrary to the very large

values of $F_{12} : PROJ$ and the conclusion of Section 5.2. Investigations of both of these components' PCA scores in the case of the CO storms showed that this peculiarity resulted from both apparently random features in the distributions of the PCA scores as well as significant sensitivity to this type in the third $F_{11}$ component. The dividing plane technique also demonstrated strong predictive power in the case of the dust samples, eliminating many of the false positives shown in Table 3, regardless of the choice of included principal components.

Figure 7 is colored by aerosol type and shows the frequency distribution of the PCA scores' distances from the pertinent separating plane. Sub-panels (a) and (c) show the distribution of biomass burning and CO storm distances against the distribution of distances for all other types. In both cases the targeted types separate clearly from the other cases. The strong separation between the biomass burning samples and other fine mode averages shown in sub-panel (b) shows that the distinguishing features of the BB cases extends significantly beyond the magnitude of the coarse mode. Sub-panel (d) shows the separation between

the AL and CO storms. The TX/OK storms are also included to illustrate how this type has many characteristics that fall in between the two other storm domains. The overlap between TX/OK storm distributions and the AL and CO storm distributions makes sense in light of the fact that these types are often confused with each other in the Mahalanobis distance based scheme (see Table 3) and have clusters that overlap in Figure 6.

## 6    Conclusions

In this work PI-Neph measurements from the DC3 and SEAC⁴RS field experiments were sub-selected and averaged over periods corresponding to stable, high quality data. An optically independent aerosol typing scheme, making use of ancillary data, was developed and the resulting 2390 cases were separated into seven classified categories, as well as an eighth unclassified category corresponding to cases that did not meet any of the classification criteria. SEAC⁴RS measurements were separated into biogenic, biomass burning, urban and dust types, based on composition measurements from the PALMS instrument, APS

data and the concentrations of various gas tracers. The geographic distribution of the resulting classification was in strong agreement with expectations suggesting a strong physical basis for the classification criteria. The DC3 dataset was divided into periods corresponding to convective system inflow over one of three storm domains located in Colorado, Texas/Oklahoma and



Alabama. 1307 cases were assigned to one of these seven categories and the remaining 1083 cases, not meeting any of the other classification criteria, were labeled as unclassified.

Phase function and $-F_{12}/F_{11}$ data are averaged over each aerosol type to obtain scattering patterns characteristic of each classification. The dust category produced a significantly stronger forward scattering peak than all other types, likely as a
result of a comparably large number of coarse mode particles. The next strongest forward peak was found in the CO storms, followed by the storms in TX/OK and then AL. These scattering patterns were also likely to be significantly influenced by coarse mode particle concentrations, with the convective systems likely lofting more large particles as the local environment becomes increasingly arid. There were very small differences in the scattering patterns of the SEAC⁴RS fine mode dominated aerosols. The largest differences between these types was observed in the peak values of $-F_{12}/F_{11}$, occurring around $90°$,
with the biomass burning cases being more strongly polarizing than the other two types.

In order to more easily explore the scattering measurements, as well as further validate the ancillary data aerosol typing scheme, principal component analysis was applied to the PI-Neph measurements and the results were examined as a function of the ancillary data classification. The first few principal components of the $F_{11}$ data and the first principal component of the $-F_{12}/F_{11}$ data showed very strong relationships with aerosol type. Two schemes were developed to divide the PCA score
space into regions representing the different classification categories. The first of these schemes was based on the Mahalanobis distances between points and the center of the cluster corresponding to each ancillary data classification category. The second scheme simply used a plane to divide the PCA score space into two regions corresponding to positive and negative classifications of a given type. In both schemes, individual scattering measurements were assigned to the correct type with very high recall, further supporting the validity of the ancillary data aerosol typing scheme, and demonstrating that the PI-Neph data alone
is capable of identifying major aerosol types. This conclusion held even in cases where differences in single angle scattering intensities between the bulk averages of two different aerosol types was very small, often much less than the error in an individual measurement. For example, the dividing plane prediction technique was able to distinguish biomass burning cases from other fine mode dominated aerosols with greater than 90% recall and TNR, despite the two categories having extremely similar average scattering patterns. The characteristics producing this clear separation between similar aerosol types are subtle and
often rely on the relationships between many angles simultaneously. This fact emphasizes the value of multi-angle scattering measurements, as well as principal component analysis's ability to reveal the underlying patterns in these datasets.

*Code and data availability.* All relevant measurements made during the SEAC⁴RS experiment are available through the SEAC4RS data archive at https://www-air.larc.nasa.gov/missions/seac4rs/ (SEAC4RS, 2013). The DC3 dataset is also available through the corresponding archive which can be found at https://www-air.larc.nasa.gov/missions/dc3-seac4rs/ (DC3, 2012). Requests for additional data can be made
to the corresponding author at reedespinosa@umbc.edu.

*Competing interests.* The authors declare that they have no conflict of interest.





*Acknowledgements.* We acknowledge funding support from the NASA Earth Science Enterprise for the SEAC$^4$RS campaign under Grant NNX12AC37G, and under the Atmospheric Composition Campaign Data Analysis and Modeling Program (ACCDAM) grant NNX14AP73G, both managed by Dr. Hal Maring. The authors would also like to thank the members of the LARGE group, particularly Bruce Anderson, Luke Ziemba and Andreas Beyersdorf for their support incorporating the PI-Neph into the LARGE instrument package. We are also grateful

5   for the scientific and technical support of the LACO team at UMBC. Additionally, we would like to thank the entire DC3 and SEAC$^4$RS science teams for providing supporting data and relevant discussion.



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



**Table 1.** The total number of cases and the number of unique flights for which at least one case was present.

| Aerosol Type | # of cases | # of flights |
|---|---|---|
| BB | 105 | 8 |
| Biogenic | 252 | 11 |
| Urban | 28 | 7 |
| Dust | 15 | 2 |
| CO Storms | 329 | 4 |
| TX/OK Storms | 535 | 5 |
| AL Storms | 140 | 2 |
| Unclassified | 986 | 17 |
| All Cases | 2390 | 37 |



**Table 2.** Recall for the Mahalanobis distance based clustering for each aerosol type, given different combinations of principal component scores. The PCA combinations that the prediction technique was applied to include: the first two $\tilde{F}_{11}$ scores ($\tilde{F}_{11} : PC_{1\text{-}2}$), the first $\tilde{F}_{11}$ and $-F_{12}/F_{11}$ scores ($\tilde{F}_{11} : PC_1; F_{12} : PC_1$), the first three $\tilde{F}_{11}$ scores ($\tilde{F}_{11} : PC_{1\text{-}3}$), the first two $\tilde{F}_{11}$ scores and first $-F_{12}/F_{11}$ scores ($\tilde{F}_{11} : PC_{1\text{-}2}; F_{12} : PC_1$) and all four scores simultaneously ($\tilde{F}_{11} : PC_{1\text{-}3}; F_{12} : PC_1$).

| Type | $\tilde{F}_{11} : PC_{1\text{-}2}$ | $\tilde{F}_{11} : PC_1; F_{12} : PC_1$ | $\tilde{F}_{11} : PC_{1\text{-}3}$ | $\tilde{F}_{11} : PC_{1\text{-}2}; F_{12} : PC_1$ | $\tilde{F}_{11} : PC_{1\text{-}3}; F_{12} : PC_1$ |
|---|---|---|---|---|---|
| Biogenic | 64.4% | 51.8% | 87.2% | 81.0% | 90.7% |
| BB | 72.4% | 29.8% | 63.8% | 84.6% | 83.7% |
| Dust | 100.0% | 100.0% | 100.0% | 100.0% | 100.0% |
| CO Storms | 72.7% | 85.1% | 76.4% | 86.2% | 86.2% |
| TX/OK Storms | 58.0% | 48.3% | 81.4% | 60.1% | 81.6% |
| AL Storms | 12.9% | 15.9% | 42.9% | 12.3% | 47.1% |





**Table 3.** Confusions matrix comparing the results of the Mahalanobis distance based prediction technique for the case where the first three $\tilde{F}_{11}$ and first $-F_{12}/F_{11}$ principal component scores are used (corresponding to the last column of Table 2) against the actual classification results of the ancillary data classification scheme. Several rows of the table sum to slightly less than the number of cases shown in Table 1 because PCA scores could not be calculated for averages containing non-physical measurements at one or more angles.

|  |  | Predicted Classification | | | | | |
|---|---|---|---|---|---|---|---|
|  |  | Biogenic | BB | Dust | CO Storms | TX/OK Storms | AL Storms |
| | Biogenic | 224 | 8 | 1 | 2 | 9 | 3 |
| | BB | 15 | 87 | 0 | 0 | 1 | 1 |
| Actual Classification | Dust | 0 | 0 | 12 | 0 | 0 | 0 |
| | CO Storms | 0 | 0 | 0 | 225 | 35 | 1 |
| | TX/OK Storms | 13 | 0 | 1 | 72 | 429 | 11 |
| | AL Storms | 52 | 5 | 0 | 2 | 14 | 65 |





**Table 4.** Recall and $TNR$ values resulting from the separating plane classification prediction technique for two different combinations of principal component scores. The value of $F_{12} : PROJ$ contains a parameter quantifying the role of the first principal component of $-F_{12}/F_{11}$ in identifying the corresponding aerosol type.

| Separated Types | $\tilde{F}_{11} : PC_{1\text{-}3}$ | | $\tilde{F}_{11} : PC_{1\text{-}2}; F_{12} : PC_1$ | | |
| --- | --- | --- | --- | --- | --- |
| | Recall | $TNR$ | Recall | $TNR$ | $F_{12} : PROJ$ |
| Dust vs. All | 94.1% | 97.6% | 93.8% | 98.3 % | 0.37 |
| BB vs. All | 90.3% | 80.3% | 90.3% | 95.0% | -0.54 |
| BB vs. Fine | 74.8% | 89.2% | 91.3% | 91.6% | -0.50 |
| CO Storms vs. All | 91.3% | 91.6% | 94.3% | 81.2% | 0.80 |
| CO vs. AL Storms | 97.6% | 95.7% | 96.2% | 97.8% | 0.83 |





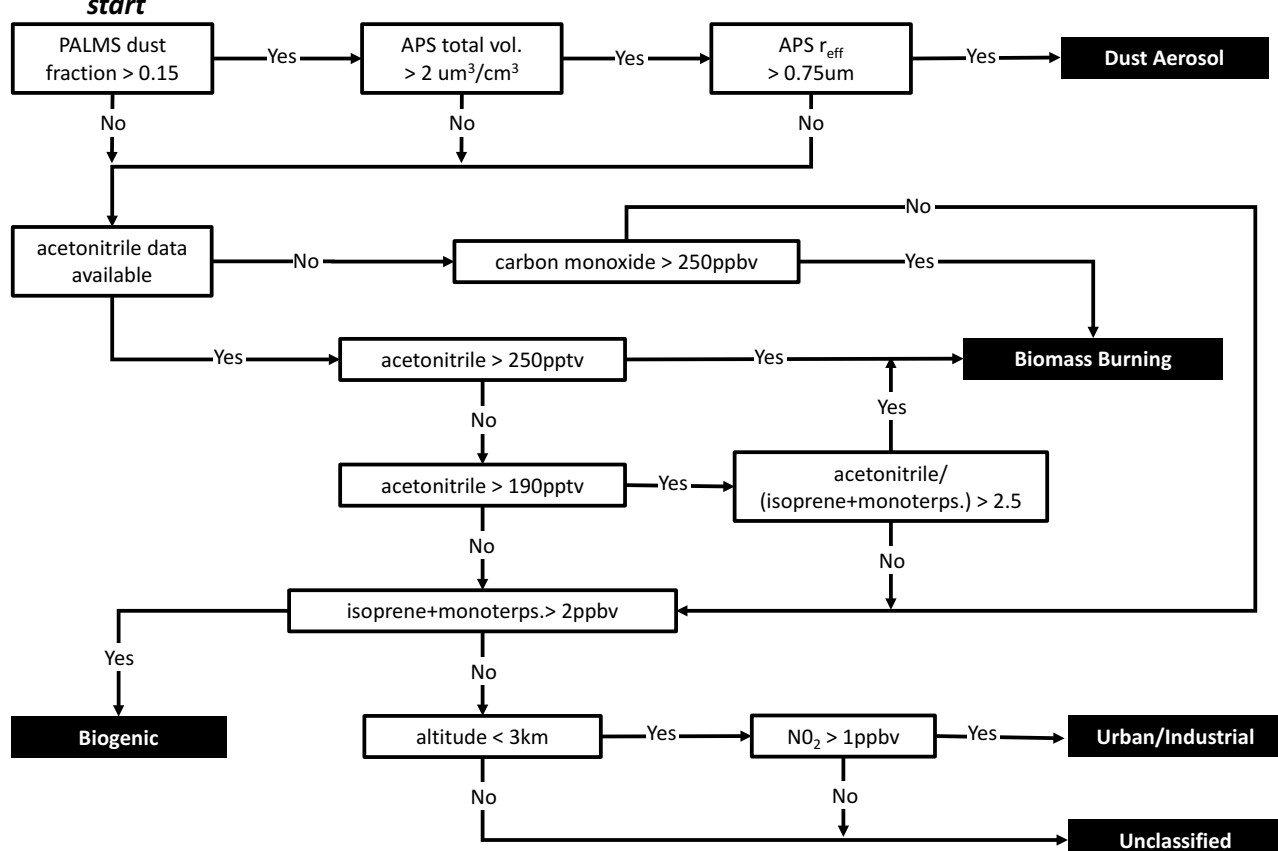

**Figure 1.** The decision tree used to classify aerosol types in SEAC[4]RS.





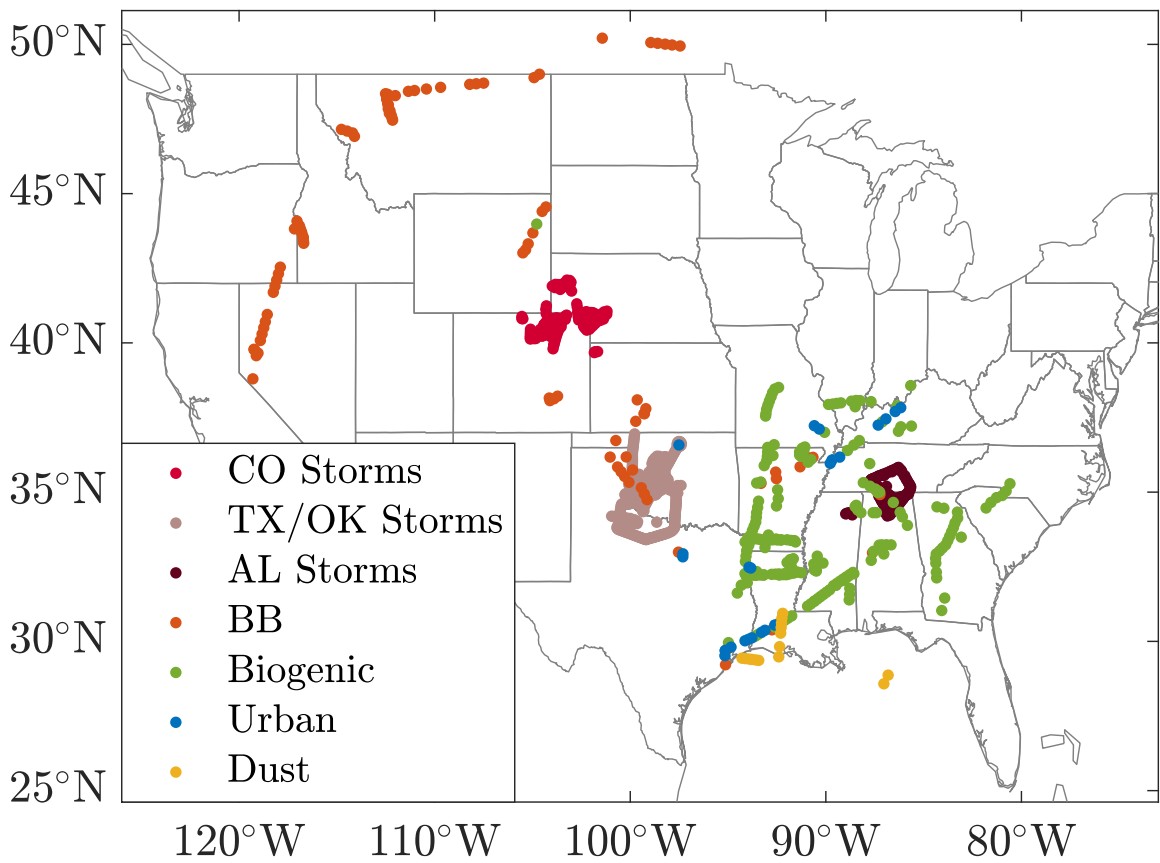

**Figure 2.** The results of the air mass classification scheme as a function of geographic location.





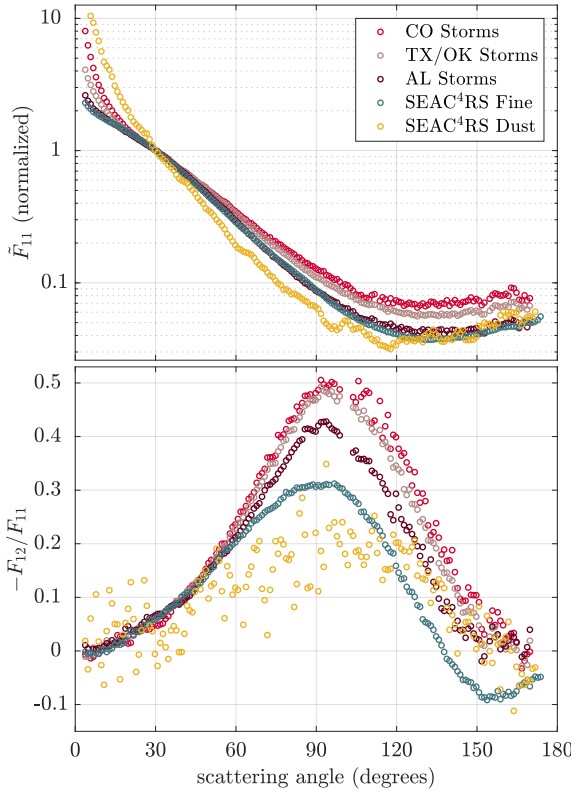

**Figure 3.** Average 532nm $\tilde{F}_{11}$ and $-F_{12}/F_{11}$ data for all three DC3 storm domains as well as dust and fine mode dominated aerosol types (biogenic, urban and BB) from SEAC$^4$RS. Normalized phase function data are scaled such that $\tilde{F}_{11}(30°) = 1$.





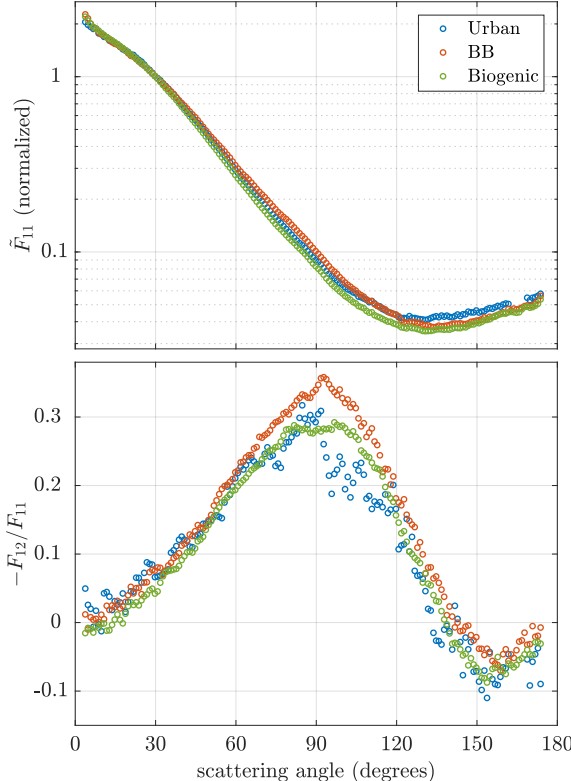

**Figure 4.** Average 532 nm $\tilde{F}_{11}$ and $-F_{12}/F_{11}$ data for the three fine mode dominated aerosol classifications in SEAC[4]RS. Small gaps in the data (ex. urban points ~165°) were removed due to strong biases from stray light.





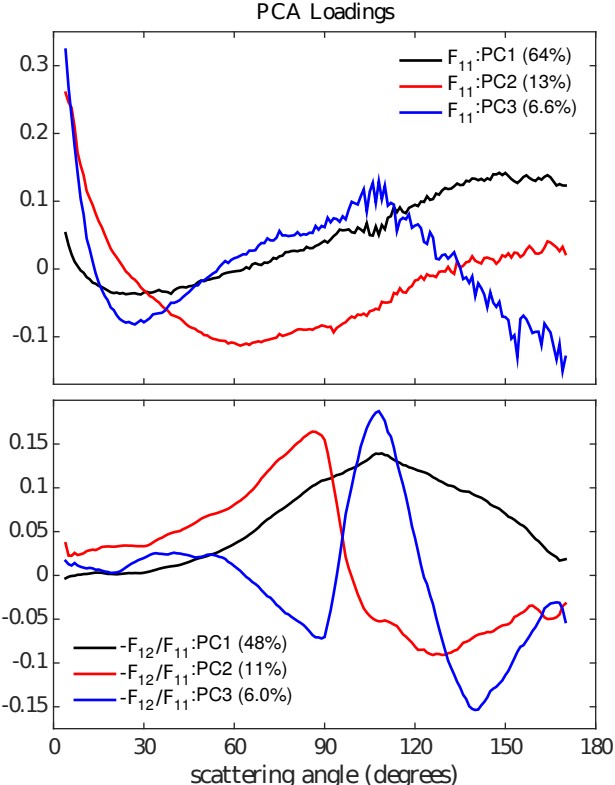

**Figure 5.** The resulting PCA loadings of the first three principal components from both the $\tilde{F}_{11}$ and $-F_{12}/F_{11}$ data. The percentage of the data's variance explained by a given component is shown in parenthesis.




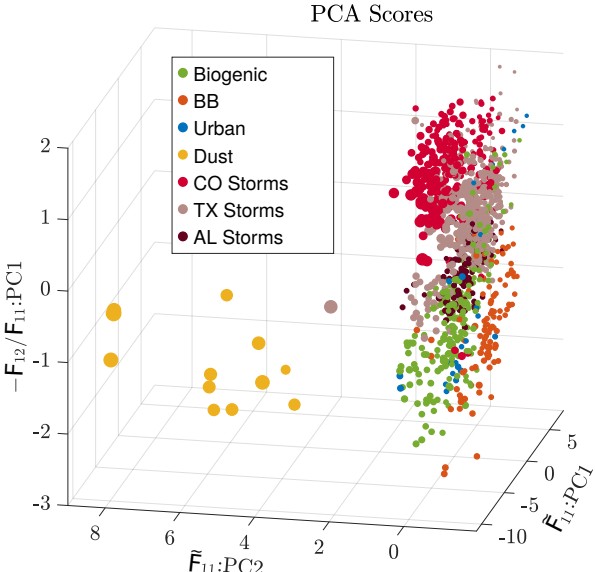

**Figure 6.** The resulting PCA scores, color coded by type, as a function of the first two $\tilde{F}_{11}$ principal component scores and the first $-F_{12}/F_{11}$ score. The points are sized according to the effective radius of the aerosol.





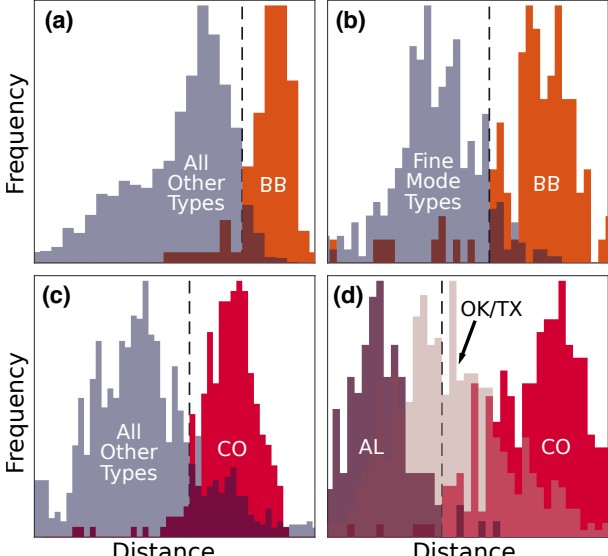

**Figure 7.** Histograms showing the separation of aerosol types along the direction perpendicular to the corresponding separating plane in the three-dimensional PCA score space composed of the first two $F_1 1$ components and first $-F_{12}/F_{11}$ component. Panel (a) shows the biomass burning cases (orange) against all other types (grey), while panel (b) shows the BB cases against only the other fine mode dominated types. Panel (c) shows the CO storms (red) against all other types, with panel (d) showing the CO storms along with only the AL (maroon) and OK/TX (beige) storms. The black vertical dashed lines represents the location of the separating plane and denotes the threshold between positive and negative classifications.