# Peer review of "In situ measurements of angular dependent light scattering by aerosols over the contiguous United States"

_Atmospheric Chemistry and Physics, 2017_

## Referee Comment (RC1) · Anonymous Referee #2 · 5 Dec 2017

The manuscript extensively describes the analysis of the data obtained by the Polarized Imaging Nephelometer (PI-Neph) during the Clouds and Climate Coupling by Regional Surveys (SEAC4RS) and the Deep Convection Clouds and Chemistry (DC3) field campaigns. This work can be divided in two well-differentiated blocks. Firstly, PI-Neph measured phase functions and degrees of linear polarization are combined with independent ancillary data classification for establishing the link between the measured scattering patterns and aerosol types classification. Secondly, it is tested whether PI-Neph light scattering data alone are sufficient for obtaining reliable aerosol types classification. It is highly appreciated the honest discussion not only of the advantages and uniqueness of the PI-Neph data but also on the systematic artifacts produced by

the instrument and potential errors in the interpretation. In total around fifty thousand raw measurements obtained during about 250 flight hours are analyzed. The paper is very well written presenting a detailed and rigorous description of the instrument, data acquisition, measurements conditions and subsequent data analysis.

I recommend publication of this paper in Atmospheric Chemistry and Physics. There are some minor issues that I would like the authors to address.

- As mentioned in Section 2.1, data corresponding to the DC3 campaign are obtained at one single wavelength (532 nm) adding to more wavelengths (473 nm and 671 nm) during the SEAC4RS campaign. By analyzing the wavelength dependence of the $-F_{12}/F_{11}$ ratio much information can be retrieved on the aerosols optical properties. However, all data presented in the paper are performed at 532 nm. There is no information/discussion on the wavelength dependence of the measured data during SEAC4RS campaign. What is the reason for that? They were finally discarded? If so, what is the reason for that?

- Section 4, third paragraph: There is a discussion about the implications on aerosols size based on the measured phase functions at back-scattering region. However, the measured phase functions are arbitrarily normalized to unity at 30 degrees. If they would be normalized to e.g. 120 degrees the AL would show the strongest back-scattering intensity. In this case it would be best to talk in terms e.g. of steepness of the phase function (measured maximum value divided by the measured minimum). Still as mentioned, the maximum of the $-F_{12}/F_{11}$ ratio is a better diagnostic tool for aerosol size specially in the fine mode peak. As stated at the end of the third paragraph the effect of particle size on the maxima of the $-F_{12}/F_{11}$ ratios is moderated by differences in the refractive index. Multiwavelength measurements of the $-F_{12}/F_{11}$ would help in disentangling both effects (size and refractive index).
* * *

---

## Referee Comment (RC2) · Anonymous Referee #4 · 13 Dec 2017

This paper presents aerosol phase function measurements made from aircraft. The phase function measurements, and other measurements, are used to investigate the possibility of classifying the aerosol types based on the measurements. The measurements are significant because they include aircraft measurements over various regions of the US. The paper presents interesting results where they are able to differentiate anthropogenic accumulation mode and soot accumulation mode aerosols using polarization measurements.

Problems: 1) to describe 1/(Mm) use (Mm)-1 not Mm-1. Please use standard scientific notation.

[Figure]

The method uses an aerosol sampler which is inside the aircraft and the air is brought in with a shrouded diffuser inlet. It is not clear what the cutoff size was for the larger aerosols. In Dolgos and Martins (2014) they state it was 5 um. This is an important issue. If it was at 5um diameter then only part of the coarse mode is being sampled. The fraction of coarse mode aerosols present in the measurement will affect the phase function (particularly in the forward scattering direction). These problem issues are not well discussed.

(pg5-35) If the aerosol are hygroscopic they can be affected by changes in RH. The authors claim the heating they are applying does not affect the measurement because they compare with the Integrating Neph which also dries the air with a nafion tube. This argument is not clear to me? The authors do mention the possibility of heating due to ram pressure but do not attempt to address this uncertainty.

---

## Referee Comment (RC3) · Anonymous Referee #3 · 15 Dec 2017

Comments for manuscript "In situ measurements of angular dependent light scattering by aerosols over the contiguous United States" by W. Reed Espinosa et all.

The manuscript describes classification of the measurements of scattering and polarized phase functions from airborne Polarized Imaging Nephelometer (PIN). The approach consist of several steps. 1. Averaging of PIN measurements over time periods suitable to reduce the effect of random noise and exclude measurements samples with very low scattering. 2. Use ancillary data to assign certain aerosol type to the given averaged sample. The ancillary data include chemical composition, aerodynamic size distribution and trace gases measurements. 3. Average measurement samples for

each aerosol type and analyze corresponding optical properties. 4. Apply Principle Component Analysis to all averaged measurement samples. PCA resulted in strong clustering of points corresponding to different aerosol types in score PCA space, which considered as the evidence of the potential of PIN observation to differentiated different aerosol types without employing of ancillary data. 5. The level of clustering by aerosol types in PCA score space is quantifies by using PCA scores only to predict the results of ancillary data classification employed two different approaches: Mahalanobis distance and dividing plane. In both schemes individual scattering measurements were assign to correct aerosol type with high recall. 6. The overall conclusion presented by authors is the PIN ability to distinguish common aerosol types without additional information. I believe that the manuscript is well in scope of ACP and can be published if below comments will be answered. 1. As seen from Fig. 6, PCA provides very good separation of dust from other aerosol types which obviously due to difference in forward scattering between coarse and fine aerosol. However, separation between fine mode aerosol types is much less distinct especially between BB and Biogenic. My guess is that the main difference between these aerosols is the different absorption level. What is the potential of PIN measurements in separating aerosols with similar particle sizes but different absorption? And if the potential is high what is the physical reason for that (PIN measures only scattering)? 2. On page 7 nm units are used along with microns. I think it is better to use the same units throughout the manuscript. 3. On Fig. 3 degree of linear polarization for SEACRS dust is much noisier than for other aerosol types. What do you think is the reason for that? 4. I am wandering if analysis in Section 4 can be supplemented with Mie or T-matrix calculations of scattering phase function and degree of linear polarization for typical (maybe AERONET based) size distributions.
* * *

---

## Referee Comment (RC4) · Anonymous Referee #1 · 18 Dec 2017

Review of In situ measurements of angular dependent light scattering by aerosols over the contiguous United States, ACP-2017-941.

General Comments

This paper is very well written and contains very unique data sets. Polar nephelometer measurements of the Aerosol Phase Function (APF) are very rare and in my opinion very lacking in atmospheric science. Various types of aerosols were measured in two aircraft campaigns with several other complimentary aerosol instruments operating simultaneously. The paper demonstrates that classifying aerosols strictly by the polar nephelometer measurements works remarkably well. This paper is appropriate

for ACP and can be posted in its discussion forum.

Specific comments Most of my comments were contained in the quick review and were addressed in this version. There are a few typos in this version.

Abstract: . . . and it is found that in most case(s) the results . . .

Page 1: Line 22: . . . the assumptions and validations of the techniques require . . .

Page2: Line 8: The limited number (of) angles . . . Line 12: . . . the results from a wide range (of) studies . . .

Page 3: Line 8: . . . required for a compressive aerosol model. Do you mean comprehensive? Line 9: .. used to measure the angular . . .

---

## Author Comment (AC1) · 8 Jan 2018

We would like to thank the reviewer #1 for his/her time and helpful suggestions. The manuscript has been updated to address each of the grammatical errors pointed out by the reviewer.

---

## Author Comment (AC2) · 8 Jan 2018

**Responses to Anonymous Referee #2**

We would like to thank the reviewer for his/her time, thoughtful insights and helpful comments. A point-by-point response to each of the reviewers concerns is listed below. The reviewers comments are shown in bold italics, while the authors' responses are indented and displayed in regular type.

***As mentioned in Section 2.1, data corresponding to the DC3 campaign are obtained at one single wavelength (532 nm) adding two more wavelengths (473 nm and 671 nm) during the SEAC4RS campaign. By analyzing the wavelength dependence of the $-F_{12}/F_{11}$ ratio much information can be retrieved on the aerosols optical properties. However, all data presented in the paper are performed at 532 nm. There is no information/discussion on the wavelength dependence of the measured data during SEAC4RS campaign. What is the reason for that? They were finally discarded? If so, what is the reason for that?***

> This manuscript focuses primarily on the ancillary data classification scheme and, more specifically, on the ability of single wavelength PI-Neph data to predict this categorization. While there certainly is significant additional information in the wavelength dependence of the scattered light the authors wanted to develop a single prediction technique that could be applied to both the DC3 and SEAC4RS dataset simultaneously. The prediction schemes presented in Sections 5.2 and 5.3 are both based on the scores derived from PCA. In order for the PCA routine to be run simultaneously on both the SEAC4RS and DC3 measurements variables that were not present in both datasets had to be discarded. As the DC3 data did not contain any measurements at 473 nm or 671 nm the corresponding SEAC4RS measurements had to be excluded.
>
> This combined dataset allows for a significantly larger number of cases on which the predictions can be made, which helps to emphasize the robustness of the technique. Furthermore, the exclusion of information contained in the spectral dependence of the scattered light helps to more clearly demonstrate the power of angular and polarization information when discerning aerosol types. It would of course be possible to simply plot the spectral dependence of some SEAC4RS types in Section 4 but the authors felt that showing this data might be a distraction from the central message of the work since these measurements were not included in the PCA based prediction schemes. Moreover, interested parties can consult Espinosa et al. (2017) which shows the spectral dependence of $F_{11}$ and $F_{12}$ measurements for several aerosols sampled during SEAC4RS. Although, it is important to note that an explicit evaluation of the additional information provided by multi-wavelength data is not performed by Espinosa et al. (2017). The authors agree that this would be a worthwhile topic of future study.

***Section 4, third paragraph: There is a discussion about the implications on aerosols size based on the measured phase functions at back-scattering region. However, the measured phase functions are arbitrarily normalized to unity at 30 degrees. If they would be normalized to e.g. 120 degrees the AL would show the strongest back- scattering intensity. In this case it would be best to talk in terms e.g. of steepness of the phase function (measured maximum value divided by the measured minimum). Still as mentioned, the maximum of the $-F_{12}/F_{11}$ ratio is a better diagnostic tool for aerosol size specially in the fine mode peak. As stated at the end of the third paragraph the effect of particle***

*size on the maxima of the $-F_{12}/F_{11}$ ratios is moderated by differences in the refractive index. Multiwavelength measurements of the $-F_{12}/F_{11}$ would help in disentangling both effects (size and refractive index).*

The authors intended to refer to the amount of light scattered within a certain angular range relative to the total amount of light scattering over all angles. This is equivalent to the value of the phase function given the alternative normalization scheme where the integral of $\tilde{F}_{11}$ over all scattering angles is set to a consistent value (ex. $4\pi$). We appreciate the reviewer pointing out the ambiguity in the original text and the first sentence of the third paragraph of Section 4 has been changed to the following:

"The same progression is evident in the backscattering angles of the DC3 storm categories, with the CO storms having the largest fraction of the total scattered light that is directed into the scattering angles larger than 90 °."

**References**

W Reed Espinosa, Lorraine A Remer, Oleg Dubovik, Luke Ziemba, Andreas Beyersdorf, Daniel Orozco, Gregory Schuster, Tatyana Lapyonok, David Fuertes, and J Vanderlei Martins. Retrievals of aerosol optical and microphysical properties from imaging polar nephelometer scattering measurements. *Atmospheric Measurement Techniques*, 10(3):811, 2017.

---

## Author Comment (AC3) · 8 Jan 2018

**Responses to Anonymous Referee #3**

We would like to thank the reviewer for his/her time, thoughtful insights and helpful comments. A point-by-point response to each of the reviewers concerns is listed below. The reviewers comments are shown in bold italics, while the authors' responses are indented and displayed in regular type.

***1. As seen from Fig. 6, PCA provides very good separation of dust from other aerosol types which obviously due to difference in forward scattering between coarse and fine aerosol. However, separation between fine mode aerosol types is much less distinct especially between BB and Biogenic. My guess is that the main difference between these aerosols is the different absorption level. What is the potential of PIN measurements in separating aerosols with similar particle sizes but different absorption? And if the potential is high what is the physical reason for that (PIN measures only scattering)?***

> The distance, within the PCA score space, between coarse mode dominated dust and other types is very significant but, as shown in Section 5.2 and 5.3 other types can be very reliably distinguished as well. For example, the "BB vs Fine" row of Table 4 shows that the biomass burning classification can be predicted from only the fine mode dominated types successfully in over 90% of the cases. In order to further address the reviewer's question, the separating plane technique was used to isolate the BB cases from only the biogenic and BB samples using the three PCA scores plotted in Figure 6. In this test it was found that the ancillary data classification could be predicted with a true positive rate of 88.5% and a true negative rate of 91.9%.
>
> The GRASP retrieval was applied to these fine mode dominated cases in accordance with the techniques of Espinosa et al. (2017) and used to explore potential differences in size and complex refractive index that may allow for the separation between BB and other fine mode dominated types. In the context of GRASP's aerosol model it was found that particle absorption and size distribution were almost identical among all of the fine mode types but real refractive index (RRI) was meaningfully elevated in the BB cases. The RRI differences alone were not sufficient to distinguish the BB particles with accuracies comparable to the PCA score based separating plane technique though so there may be other factors (ex. particle morphologies that aren't included in GRASP's aerosol model) playing a role as well.
>
> While the GRASP/PI-Neph retrieval does not have high sensitivity to absorption some sensitivity does exist due to unique changes in the angular dependence of the scattering intensities that are associated with changes in the imaginary part of the refractive index. The results of the GRASP/PI-Neph retrieval have been compared with absorption measurements made in parallel during DC3 by a Particle Soot/Absorption Photometer (PSAP). When GRASP is applied to PI-Neph light scattering measurements alone a correlation ($R \approx 0.4$) is found between the measured and retrieved absorption coefficients but the retrieved values are biased high by more than a factor of two. This comparison is part of an on going effort by the authors to better understand absorption's impact on the parameters retrieved using the methods of Espinosa et al. (2017).

***2. On page 7 nm units are used along with microns. I think it is better to use the same units throughout the manuscript.***

> All references to particle size have been converted to $\mu$m.

***3. On Fig. 3 degree of linear polarization for SEACRS dust is much noisier than for other aerosol types. What do you think is the reason for that?***

> The noise in the dust measurements is primarily driven by the high dynamic range of the corresponding phase functions. Please see the brief discussion of this effect around line 10 of page 10 of the original discussion paper.

***4. I am wandering if analysis in Section 4 can be supplemented with Mie or T-matrix calculations of scattering phase function and degree of linear polarization for typical (maybe AERONET based) size distributions.***

> The GRASP retrieval (Dubovik et al., 2011; Espinosa et al., 2017), which includes a spheroid model, as well as a separate Mie code (Mishchenko et al., 2002) was used to explore the relationships between size and refractive index of the particles and the scattering properties that are discussed in Section 4. Multiple changes have been made to the text to clarify this point.
>
> The sentence beginning on line 32 of page 9 of the original discussion paper has been expanded to:
> "Mie computations (Mishchenko et al., 2002) were performed in order to identify the size, shape and complex refractive index changes that are potentially driving this progression. In size distribution and refractive index regimes typical of ambient, fine mode aerosol ($r_v \approx$150 $\mu$m; $\sigma$ =0.38; $m$ =1.5+0.01$i$) the backscattering region of the phase function is very sensitive to the diameter of the particles, suggesting significantly smaller fine mode particles in the CO inflow than in the AL inflow."
>
> Additionally, the sentence beginning on line 5 of page 10 of the original discussion paper has been changed to:
> "Mie simulations and inversions of the angular scattering measurements (Dubovik et al., 2011, 2014; Espinosa et al., 2017) suggest that the reduced $-F_{12}/F_{11}$ maximum observed in this fine mode dominated data is likely driven primarily by fine mode particles with slightly larger diameters than those found in the three DC3 categories."
>
> Lastly, a reference discussing the relationship between the forward scattering peak and fraction of coarse mode particles (Russell et al., 2004) has been added to Section 4.

**References**

O. Dubovik, M. Herman, a. Holdak, T. Lapyonok, D. Tanré, J. L. Deuzé, F. Ducos, a. Sinyuk, and a. Lopatin. Statistically optimized inversion algorithm for enhanced retrieval of aerosol properties from spectral multi-angle polarimetric satellite observations. *Atmospheric Measurement Techniques*, 4(5):975–1018, 2011. ISSN 1867-8548. doi: 10.5194/amt-4-975-2011.

Oleg Dubovik, Tatyana Lapyonok, Pavel Litvinov, Maurice Herman, David Fuertes, Fabrice Ducos, Benjamin Torres, Yevgeny Derimian, Xin Huang, Anton Lopatin, Anatoli Chaikovsky, Michael Aspetsberger, and Christian Federspiel. GRASP: a versatile algorithm for characterizing the atmosphere. *SPIE Newsroom*, pages 2–5, 2014. ISSN 18182259. doi: 10.1117/2.1201408.005558.

W Reed Espinosa, Lorraine A Remer, Oleg Dubovik, Luke Ziemba, Andreas Beyersdorf, Daniel Orozco, Gregory Schuster, Tatyana Lapyonok, David Fuertes, and J Vanderlei

Martins. Retrievals of aerosol optical and microphysical properties from imaging polar nephelometer scattering measurements. *Atmospheric Measurement Techniques*, 10(3):811, 2017.

Michael I Mishchenko, Larry D Travis, and Andrew A Lacis. *Scattering, absorption, and emission of light by small particles.* Cambridge university press, 2002.

P. B. Russell, J. M. Livingston, O. Dubovik, S. A. Ramirez, J. Wang, J. Redemann, B. Schmid, M. Box, and B. N. Holben. Sunlight transmission through desert dust and marine aerosols: Diffuse light corrections to sun photometry and pyrheliometry. *Journal of Geophysical Research: Atmospheres*, 109(D8):n/a–n/a, 2004. ISSN 2156-2202. doi: 10.1029/2003JD004292. URL `http://dx.doi.org/10.1029/2003JD004292`. D08207.

---

## Author Comment (AC4) · 8 Jan 2018

**Responses to Anonymous Referee #4**

We would like to thank the reviewer for his/her time, thoughtful insights and helpful comments. A point-by-point response to each of the reviewers concerns is listed below. The reviewers comments are shown in bold italics, while the authors' responses are indented and displayed in regular type.

***To describe 1/(Mm) use $(Mm)^{-1}$ not $Mm^{-1}$. Please use standard scientific notation.***

> The reciprocal milliseconds example in rule 5 of the NIST style guide for SI units (https://physics.nist.gov/cuu/Units/checklist.html) suggests that $Mm^{-1}$ is also an acceptable form. Additionally, in a brief survey of papers recently published in ACP (Zhang et al., 2015; Rosati et al., 2016; Bi et al., 2017) we found the form $Mm^{-1}$ to be used in all instances. For these reasons, the authors feel that using $Mm^{-1}$ will maximize clarity for the reader but we are certainly open to changing to the $(Mm)^{-1}$ form if the reviewer or the editor strongly favors doing so.

***The method uses an aerosol sampler which is inside the aircraft and the air is brought in with a shrouded diffuser inlet. It is not clear what the cutoff size was for the larger aerosols. In Dolgos and Martins (2014) they state it was 5 um. This is an important issue. If it was at 5um diameter then only part of the coarse mode is being sampled. The fraction of coarse mode aerosols present in the measurement will affect the phase function (particularly in the forward scattering direction). These problem issues are not well discussed.***

> The authors appreciate the reviewer catching the ommision of this information. The inlet is known to have a 50% passing efficiency at an aerodynamic radius of $1.8\mu m$ (McNaughton et al., 2007). This statistic has been added to the fourth paragraph of section 2.2.
>
> Additionally, the following sentence has been added to the end of the fourth paragraph of Section 4:
> "It should also be noted that the true forward scattering peak of the ambient aerosol may be even larger than the values reported by the PI-Neph, whose sample is subject to inlet cutoff effects which disproportionately effect the largest particles."

***(pg5-35) If the aerosol are hygroscopic they can be affected by changes in RH. The authors claim the heating they are applying does not affect the measurement because they compare with the Integrating Neph which also dries the air with a nafion tube. This argument is not clear to me? The authors do mention the possibility of heating due to ram pressure but do not attempt to address this uncertainty.***

> The aerosol data reported here are measured under predominantly dry conditions and are not intended to represent the scattering properties at ambient relative humidities. The comparison with the Integrating Nephelometer measurements is intended to show that the potential evaporation of volatile compounds, resulting from our temperature based drying procedure, has very little effect on the scattering properties of the dry particles.

In order to help clarify this point the first sentence of the paragraph in question was replaced with the following text:

"The scattering properties of hygroscopic aerosols are influenced by the uptake of water which typically occurs at relative humidities (RH) greater than 40% (Ziemba et al., 2013; Orozco et al., 2016). The PI-Neph's sample was conditioned with a temperature-controlled drier that reduced the sample's RH by heating the incoming ambient air to a temperature of 35°C. In almost all cases this approach was found to reduce the sample's RH below 40% so the reported properties are thought to be representative of "dry" particles.

**References**

Jianrong Bi, Jianping Huang, Jinsen Shi, Zhiyuan Hu, Tian Zhou, Guolong Zhang, Zhong-wei Huang, Xin Wang, and Hongchun Jin. Measurement of scattering and absorption properties of dust aerosol in a gobi farmland region of northwestern china–a potential anthropogenic influence. *Atmospheric Chemistry and Physics*, 17(12):7775, 2017.

Cameron S. McNaughton, Antony D. Clarke, Steven G. Howell, Mitchell Pinkerton, Bruce Anderson, Lee Thornhill, Charlie Hudgins, Edward Winstead, Jack E. Dibb, Eric Scheuer, and Hal Maring. Results from the DC-8 Inlet Characterization Experiment (DICE): Airborne Versus Surface Sampling of Mineral Dust and Sea Salt Aerosols. *Aerosol Science and Technology*, 41(2):136–159, 2007. ISSN 0278-6826. doi: 10.1080/02786820601118406.

Daniel Orozco, A. J. Beyersdorf, L. D. Ziemba, T. Berkoff, Q. Zhang, R. Delgado, C. J. Hennigan, K.L. Thornhill, D. E. Young, C. Parworth, H. Kim, and R. M. Hoff. Hygroscopicity Measurements of Aerosol Particles in the San Joaquin Valley, CA, Baltimore, MD, and Golden, CO. *Journal of Geophysical Research: Atmospheres*, 121:7344–7359, 2016. ISSN 2169897X. doi: 10.1002/2015JD023971.

Bernadette Rosati, Erik Herrmann, Silvia Bucci, Federico Fierli, Francesco Cairo, Martin Gysel, Ralf Tillmann, Johannes Größ, Gian Paolo Gobbi, Luca Di Liberto, et al. Studying the vertical aerosol extinction coefficient by comparing in situ airborne data and elastic backscatter lidar. *Atmospheric Chemistry and Physics*, 16(7):4539–4554, 2016.

L Zhang, JY Sun, XJ Shen, YM Zhang, H Che, QL Ma, YW Zhang, XY Zhang, and JA Ogren. Observations of relative humidity effects on aerosol light scattering in the yangtze river delta of china. *Atmospheric Chemistry and Physics*, 15(14):8439–8454, 2015.

Luke D. Ziemba, K. Lee Thornhill, Rich Ferrare, John Barrick, Andreas J. Beyersdorf, Gao Chen, Suzanne N. Crumeyrolle, John Hair, Chris Hostetler, Charlie Hudgins, Michael Obland, Raymond Rogers, Amy Jo Scarino, Edward L. Winstead, and Bruce E. Anderson. Airborne observations of aerosol extinction by in situ and remote-sensing techniques: Evaluation of particle hygroscopicity. *Geophysical Research Letters*, 40(2):417–422, 2013. ISSN 00948276. doi: 10.1029/2012GL054428.

---

## Author Response (AR2)

**Following from the last comment of reviewer #4, the editor has expressed concern that RH<40% may not be sufficiently low to guarantee dry particles. The authors are sympathetic to this concern and have softened the language of text accordingly. The relevant paragraph of Section 2.2 now reads:**

[revised manuscript text omitted]